# Risk Factors for Early Implant Failure and Selection of Bone Grafting Materials for Various Bone Augmentation Procedures: A Narrative Review

**DOI:** 10.3390/bioengineering11020192

**Published:** 2024-02-17

**Authors:** Motohiro Munakata, Yu Kataoka, Kikue Yamaguchi, Minoru Sanda

**Affiliations:** 1Department of Implant Dentistry, Showa University School of Dentistry, 2-1-1, Kita-senzoku, Ota-ku, Tokyo 1458515, Japan; 2Department of Dental Education, Showa University School of Dentistry, 1-8-5, Hatanodai, Shinagawa-ku, Tokyo 1428555, Japan; 3Department of Biomaterials and Engineering, Showa University School of Dentistry, 1-8-5, Hatanodai, Shinagawa-ku, Tokyo 1428555, Japan; 4Department of Prosthodontics, Showa University School of Dentistry, 2-1-1, Kita-senzoku, Ota-ku, Tokyo 1458515, Japan

**Keywords:** bone graft material, alveolar ridge augmentation, dental implant therapy, sinus floor augmentations, bone substitute

## Abstract

Implant therapy is now an established treatment with high long-term success and survival rates. However, early implant failure, which occurs within one year of superstructure placement, occurs at a higher rate than late failure, which is represented by peri-implantitis caused by bacterial infection. Furthermore, various risk factors for early failure have been reported, including patient-related factors, such as systemic diseases, smoking, and bone quality and quantity, as well as surgery-related factors, such as surgeons’ skill, osteogenesis technique, and selection of graft material, and implant-related factors, such as initial implant fixation and implant length diameter. Due to the wide variety of relevant factors reported, it is difficult to identify the cause of the problem. The purpose of this review is to discuss the risk factors associated with various types of bone augmentation which have a close causal relationship with early implant failure, and to determine the optimal bone grafting material for bone augmentation procedures to avoid early implant failure.

## 1. Introduction

Dental implant therapy is now established as a stable long-term treatment method, and has shown a high survival rate [1,2,3]. However, in a large-scale clinical study conducted by Derks et al. [4] on the prognosis of implant therapy in a Swedish population, the overall implant loss rate over a 9-year observation period was reported to be as low as 7.6%; however, implant loss occurring before the placement of the superstructure was 4.4%. Moreover, Lin et al. [5] demonstrated a high survival rate both at the patient (98.0%) and implant (98.7%) levels in a study evaluating early and late implant failure in 18,199 patients with 30,959 implants. Among these, 194 (0.6%) implants were lost before abutment connection, and 209 (0.7%) implants were lost during the 6-year observation period after abutment connection. Notably, early failures within the first 6 months after surgery accounted for 48.1% of all lost implants over the entire study period. As evidenced by these reports, early failure in dental implant treatment is caused by risk factors different from those affecting long-term survival rates. Therefore, knowledge of the causes of early implant failure is more important than the survival rate in terms of short-term prognosis, which is also extremely important in terms of recovery and building trust between dentists and patients. In this review, we discuss the causes of early implant failure, risk factors for various bone augmentation procedures, and the selection of appropriate graft material to avoid early implant failure.

## 2. Early Implant Failure 

### 2.1. Difference between Early Failure and Late Implant Failure

Implant loss can be categorised into early and late failure. Late failure can be observed in cases of peri-implantitis, which are primarily attributed to bacterial infection. Other causes include factors such as excessive occlusal forces, bruxism, and design challenges in the superstructure, including the number of implants and cantilevers, resulting in implant fracture or disintegration. Late implant failure could be attributed to bacterial infection owing to poor plaque control, abfraction caused by cantilevers or bruxism, and overload due to an insufficient number of implants or the effects of the diameter and length of the implant [6,7,8,9,10,11]. However, the number of risk indicators for late implant failure is surprisingly small, and corresponding measures and preventive strategies have been increasingly elucidated (Figure 1).

On the other hand, early failure, while categorised by researchers into distinct periods, such as ① implant placement to superstructure placement, ② implant placement to 6 months post-implantation, and ③ implant placement to 1 year after superstructure placement, lacks a clear and universal definition or criteria. Moreover, unlike late failure, no precise definition exists for risk indicators associated with early failure. Various factors, including systemic diseases, smoking, bone quantity and quality, implant site and surgical technique, implant diameter and length, implant torque (initial stabilisation), selection of graft material, and the skill of the surgeon, encompassing patient-, implant-, and surgeon-related factors, have been reported as potential risk indicators [12,13,14,15]. The multitude of reported factors makes it highly challenging to pinpoint the specific causes of early failure.

### 2.2. Problems Associated with Early Implant Failure

Prompt recovery by means of a second surgery is warranted following early implant failure. However, Zhou et al. [16] conducted a systematic review highlighting the challenges associated with a second surgery (reimplantation) at the site of implant loss and reported a survival rate of 88.8% (mean observation period of 41.5 months) for reimplantation; moreover, the survival rate for a second reimplantation was notably lower at 74.2% (mean observation period of 29.7 months). They identified smoking and poor oral hygiene status, (vertical and horizontal) bone quantity, and soft tissue conditions around the implant as risk factors. Park et al. [17] investigated the survival rate and risk factors for reimplantation; they reported an implant failure rate of 11.6% for reimplantation, which was predominant in cases involving the upper jaw or bone augmentation. Notably, all implant failures associated with reimplantation were identified as early failures. Additionally, Agari et al. [18] conducted a similar study and demonstrated that, while the survival rate for the initial surgery was 95.4%, the survival rates for the first reimplantation (reoperation), second reimplantation (second reoperation), and third reimplantation (third reoperation) were 77.4%, 72.7%, and 50.0%, respectively. They reported that the majority of reimplantations were associated with early failure. Furthermore, Malo et al. [19] reported that sites with previous implant failure were at the highest risk for both peri-implant disease and bone resorption (marginal bone loss (MBL) > 3 mm) in a study investigating the risk factors for the all-on-4 treatment in the mandible with a long-term follow-up of 10 to 18 years. These reports collectively emphasise that reimplantation is the most challenging surgery among all implant-related procedures (Table 1 reports the definition and risk factors for early implant failure).

### 2.3. Risk Factors for Early Implant Failure

Jemt et al. [32] found that early implant failure occurred in 11.3% of the edentulous patients in a study on implant failure conducted over a 15-year period; 71.8% of these early failures occurred up to the point of superstructure placement, which suggests that the majority of early implant failures occur before the placement of the superstructure. Additionally, Carr et al. [37] reported a failure rate of 4.2% within the first year post-implantation, which was the highest within the entire observation period, with a mean failure period of 129 days, indicating a very short time frame for early failure. These reports highlight that early implant failure should be considered as a mechanism separate from the long-term survival rate. As these reports indicate, early implant failure should be viewed as a risk factor for a completely different mechanism than that of the long-term survival rate, and the risk factors related to the “acquisition of bone union” should be evaluated. 

As shown in Table 1, risk factors for early implant failure have been reported in many papers, including systematic reviews. The risk factors cited, including in the latest report by da Rocha Costa Coelho et al. [38], include ① patient-related factors, such as sex, age, systemic diseases, smoking, regular medication, bone quantity and quality, implantation site (maxilla, mandible, premolar, molar), and defect type (dentulous or edentulous); ② implant-related factors, such as implant diameter and length, implant manufacturer (surface characteristics), implant design [39,40], titanium ion leakage [41], and placement torque; and ③ surgeon-related factors, such as bone augmentation, surgeons’ skill and experience, postoperative complication, graft material, and patient management (administration of antibacterial agents and use of postoperative dentures). 

However, the definition of early failure differs among researchers, and the time period of the studies on implant placement (1980s–2010s), which is related to the surface characteristics, shape, and technique of the implants, varies; thus, no consensus on risk factors has been reached. 

## 3. Relationship between Early Implant Failure and the Bone Augmentation Procedure

As observed in Table 1, many of the factors listed as risk factors for early implant failure in clinical studies are not systemic diseases, such as diabetes and osteoporosis, nor factors related to bone quality, such as the placement site, initial stabilisation (placement torque), implant diameter and length, plaque control of the patient, or periodontal disease of the remaining teeth, which are taken into consideration by dentists when placing implants, but are instead factors related to bone augmentation (technique and graft material). The mechanism is completely different from that of late failure of implants regarding long-term prognosis. This is because the late failure of implants is the so-called ‘loss of osseointegration’ caused by peri-implantitis that results in MBL or parafunctions that result in disintegration, whereas early failure is the ‘failure to achieve osseointegration’ that is also caused by postoperative infection. Therefore, preventing failure to achieve osseointegration can prevent early implant failure.

Upon extracting clinical studies on the relationship between early implant failure and bone augmentation surgery, Antoun et al. [30], who focused on early failure (between placement to 1 year of loading) in 1592 implants, found that over 75% of the implants failed within two months after loading. They identified smoking habits, surgical skills (number of years of experience), and the use of GBR and immediate or one-stage procedures as risk factors for early failure. Similarly, Jemt [29] conducted a study on the risk factors for early implant failure (within the first year after placement) in 3448 cases using the same implant system. The study reported that the surgeon’s skill (hazards ratio (HR): 5.13) was the most significant factor, emphasising that early implant failure, including sinus lift, GBR, and immediate extraction, is dependent on the surgeon. Carr et al. [37] conducted a study on the risk factors for early implant failure (within the first year after placement) in 8540 implants of 362 cases. They considered various factors, including patient-, implant-, and surgeon-related factors. The study revealed that 87.8% of the implant failures were attributed to the surgical techniques, and a strong association was observed between early implant failure and bone grafting alone (HR: 1.45), ridge preservation (HR: 2.67), xenografts (HR: 2.12), and complications arising from surgical procedures (HR: 15.84). Kang et al. [24] investigated early implant failure (before and within a few weeks after the superstructure placement) in 409 cases with 1031 single platform-switched bone-level implants; they reported that the surgeon’s number of years of experience (skill level) was the most significant factor, and implants with GBR (simultaneous or delayed) or lateral sinus lift showed a significantly higher incidence of early implant failure. Chang [14] conducted a study on early implant failure (from implantation to just before the final placement of the superstructure) in 1050 implants of 376 cases, and identified bone graft as the most significant risk factor for early failure in the maxilla, with an odds ratio (OR) of 9.45, followed by postoperative inflammatory symptoms as high-risk factors, with ORs of 3.47 and 6.69 in the maxilla and mandible, respectively. In a study by Yang et al. [18] comprising 1078 cases with 2053 implants investigating early implant failure (from implantation to within one year of loading), Type I bone quality (OR: 3.689), bone augmentation (OR: 1.742), and immediate implant placement (IIP) after tooth extraction (OR: 3.509) were identified as risk factors. Tattan et al. [33] also identified the risk factors for early implant failure (before placement of the prosthesis). They did not find smoking, diabetes, systemic diseases, such as osteoporosis, history of periodontal disease, and implant length or design (tissue level/bone level) to be risk factors; however, instead, surgical procedures, such as socket preservation (HR: 7.5), simultaneous soft tissue grafting during implant placement (HR: 5.03), and simultaneous bone grafting (HR: 3.4), were identified as risk factors. In a study by Wu et al. (2021) [21] involving 3785 cases and 6113 implants, an investigation into early implant failure (before the placement of the prosthesis) reported an early failure rate of 1.6% at the patient level and 1.2% at the implant level. The identified risk factors included the maxillary molar region (OR: 2.73), implant surface characteristics, and the site of bone grafting. In particular, staged approach bone grafting in the maxilla and mandible and male sex were considered risk factors. Additionally, Clauser et al. [42] conducted a systematic review and meta-analysis on the association between bone grafts and early implant failure and concluded that there is a significant association between bone augmentation procedures and early implant failure (OR: 1.50), as well as that bone augmentation procedures may have a negative impact on implant osseointegration. As shown above, the bone augmentation techniques and graft materials, including the skill and experience of the surgeon and postoperative complications, are important factors associated with ‘failure to achieve osseointegration’, which is the cause of early implant failure. Thus, graft material selection for various surgical procedures and appropriate surgical techniques can prevent early implant failure.

## 4. Risk Factors for Early Implant Failure and Selection of Graft Material in Various Surgical Procedures

### 4.1. Alveolar Ridge Preservation

#### 4.1.1. Efficacy of Alveolar Ridge Preservation and Associated Complications 

Alveolar ridge preservation (ARP) reportedly results in favourable outcomes for the maintenance of both the width and height of the alveolar ridge compared to simple tooth extraction, as revealed by systematic reviews and meta-analyses [43,44,45]. Furthermore, studies have reported variations in the alveolar ridge width and height resorption based on the technique of the ARP (flap or flapless, and open or closed wound) [46]. In recent research, Avila-Ortiz et al. [47] performed ARP in cases where the thickness of the labial bone wall was less than 1 mm on average. They compared these with cases of simple tooth extraction and found that ARP significantly suppressed bone reduction in both the horizontal and vertical dimensions of the alveolar ridge. Additionally, they reported that ARP could suppress alveolar ridge resorption by at least 10% in three-dimensional bone volume (mm^3^). Moreover, Cha et al. [48] examined cases of ARP in the upper molars, anticipating sinus augmentation. They revealed that the existing bone volume was significantly greater with ARP (7.30 mm) compared to extraction alone (4.83 mm) at 6 months following tooth extraction. Moreover, in cases treated with ARP, implant placement could be completed without sinus augmentation in 42.9% of cases, while all cases treated with extraction alone needed sinus augmentation, thereby suggesting the effectiveness of ARP in both the anterior and molar region (Figure 2).

Furthermore, considering bone defects in extraction sockets, Lee et al. [49] investigated the impact of ARP on implant placement in complex extraction socket bone defects resulting from periodontitis or endo-perio lesions; implant placement was challenging in 4.7% of cases in the group without ARP compared with 0.8% of cases in the group with ARP. Additionally, ARP demonstrated a significantly lower overall need for bone augmentation (45.0% vs. 23.5%), horizontal bone augmentation procedures (38.6% vs. 18.2%), vertical bone augmentation procedures (15.2% vs. 1.2%), and sinus augmentation (15.8% vs. 6.5%), thereby indicating the effectiveness of ARP. A systematic review by Atieh et al. [50] also reported ARP to be highly effective even in cases with severe bone defects, leading to a significant reduction in the need for additional bone grafts during implant placement. This highlights that ARP is an excellent technique for suppressing alveolar bone resorption and minimising bone grafting during implant placement.

However, numerous reports have addressed complications associated with ARP. Early complications within the first two weeks of the procedure include infection (2.8–9.1%) and dehiscence (2.6%); additionally, persistent inflammatory symptoms include redness, swelling, and bleeding lasting more than two weeks, as well as complications related to the technique itself, involving graft material leakage or loss (2.9–14.3%), exposure of the membrane due to primary closure, mucosal perforation, and loss of keratinised mucosa. Complications related to implant placement include poor initial stability during placement due to the fragility of the transplanted bone, and the need for regrafting procedures due to loss of the bone graft material.

Furthermore, regarding the association of ARP with early implant failure, Hoang and Mealey [51] reported an early implant failure rate of 3.3%, while Lee et al. [33] reported rates of 1.6% and 0.6% (with and without ARP, respectively), indicating that ARP may not always provide favourable outcomes. Therefore, ARP should not be considered as a crude procedure of merely ‘filling the extraction socket with graft material’.

#### 4.1.2. Selection of Bone Grafting Materials in ARP

In the systematic review and meta-analysis by De Risi et al. [52] regarding ARP graft materials, the residual bone graft had the lowest survival rate when using allografts (12.4–21.1%), whereas xenografts and alloplasts had the lowest survival rates after 7 months; however, they later demonstrated high survival rates of 37.1% and 37.2%, respectively. Additionally, concerning the bone ratio, allografts showed a high ratio of 54.4% at 3 months, while xenografts demonstrated the lowest ratio of 23.6% after 5 months. Graft materials can reportedly lead to the delayed formation of new bone and reduced bone contact rate. Santana et al. [53] histologically and radiographically evaluated ARP using allograft, xenograft, and blood clots; the radiographic evaluation at 6 months indicated superiority of the allograft and similar findings with the blood clot and xenograft; however, in terms of the rate of new bone formation (NBF), the blood clot (47.8%) showed greater NBF followed by allografts (33.3%) and xenografts (28.2%), thereby suggesting that allografts are more effective than xenografts.

Furthermore, the systematic review by Jambhekar et al. [54] revealed horizontal bone loss without graft material to be 2.79 mm; xenograft had the smallest amount of bone loss at 1.3 mm, followed by allograft (1.63 mm) and alloplast (2.13 mm). Although xenograft showed favourable clinical outcomes, alloplast had the highest rate of NBF at 45.5%, followed by no graft material at 41.1%, xenograft at 35.7%, and allograft at 29.9%. Moreover, the survival rate was the highest for allograft at 21.8%, followed by xenograft at 19.3%, and alloplast at 13.7%, which is contrary to the clinical outcomes. Thus, the selection of a non-absorbable graft material for ARP may reduce bone contact rates during implant placement, potentially resulting in a higher risk of early implant failure. Furthermore, in the latest systematic review by Corbella et al. [55] on histological studies of various graft materials (no graft material, bovine bone, allograft, porcine bone, hydroxyapatite (HA), beta-tricalcium phosphate (β-TCP)) in ARP, no significant difference was observed in bone formation among the graft materials. However, they drew the following conclusions: ① calcium sulphate and β-TCP resorbed more rapidly compared to other graft materials, while xenografts had a slower rate of resorption than allografts; ② bovine bone demonstrated significantly less new bone volume compared to naturally healed sites, whereas porcine bone and HA showed greater new bone volume; and ③ allografts did not show a significant difference in new bone volume compared to naturally healed sites. Thus, although non-resorbable xenografts (especially bovine bone) may provide a clinical perception of being replaced by bone, they may persist only as remnants.

For example, graft materials, including xenografts or alloplasts such as HA, generally have a slow resorption rate and may persist even after seven months post-transplantation. Thus, it may be advisable to extend the timing of implant placement or the unloaded period to ensure proper bone contact. On the other hand, in the case of allografts or artificial bones, such as β-TCP, which have a fast resorption rate three months after transplantation, early implant placement should be considered. However, subsequent jawbone resorption may occur owing to the rapid resorption.

Couso-Queiruga et al. [56] investigated the impact of the healing period on the bone quality of graft materials using bovine bone + collagen. They divided participants into three groups based on the post-extraction periods (3, 6, and 9 months) and observed the changes in the graft material. The results revealed that the percentage of NBF increased (13.5%, 33.3%, 37.1%, respectively), while the percentage of residual bone graft decreased (16.9%, 10.7%, 9.5%, respectively) in the group with a longer post-extraction period. However, they reported that greater resorption was observed in both the hard and soft tissues of the jawbone in the groups with a longer post-extraction period.

Thus, ARP is an effective technique for controlling horizontal and vertical alveolar ridge resorption; however, selecting an appropriate graft material and establishing the healing period are crucial for preventing early implant failure.

### 4.2. Alveolar Ridge Augmentation (Vertical/Horizontal) 

#### 4.2.1. Relationship between Alveolar Ridge Augmentation, Complications, and Early Failure

A systematic review and meta-analysis by Lim et al. [57] regarding the complications related to alveolar ridge augmentation concluded that complications, including membrane exposure, soft tissue dehiscence, and postoperative infections, occur at a very high rate of 16.8%, thus highlighting that alveolar ridge augmentation is a technically sensitive procedure with a significant potential for complications, often influenced by the skill of the surgeon. Moreover, the association between bone augmentation and early implant failure (OR: 1.50) and the negative impact of bone augmentation on the osseointegration of the implant, as concluded by the systematic review by Clauser et al. [26] and clinical studies on early implant failure by Chang [27] and Lin et al. [5] where bone grafting was identified as a high-risk factor with respective ORs of 9.45 and 1.29, underscores that GBR and alveolar ridge augmentation, such as horizontal or vertical bone augmentation procedures, are significant risk factors for early implant failure, including postoperative complications (Figure 3). 

Jensen et al. [58] evaluated bone augmentation in a study involving 223 individuals and 350 implants. They reported that soft tissue dehiscence occurred in 1.7% of cases after GBR, 25.9% after horizontal bone augmentation (staged), and 18.2% after vertical bone augmentation (staged). Postoperative infections were observed in 2%, 11%, and 9% of GBR, horizontal bone augmentation (staged), and vertical bone augmentation (staged) cases, respectively. The occurrence of early implant failure was 1.7% (six implants), with four implants associated with GBR (1.6%) and two implants with vertical bone augmentation (staged) (12%). Thus, vertical bone augmentation, in particular, is a possible risk factor for early implant failure. 

#### 4.2.2. Selection of Bone Graft Material for Alveolar Ridge Augmentation

Troeltzsch et al. [59] conducted a systematic review of the effects of graft materials and membranes in horizontal and vertical bone augmentation. The overall defect fill rate was reportedly 79.8 ± 18.7%. Graft materials showed results ranging from 51.0 ± 13.6% (alloplast) to 85.8 ± 13.4% (xenograft). Considering horizontal bone augmentation, the average gain was 3.7 ± 1.2 mm, whereas the average gain was 4.5 ± 1.0 mm for autogenous bone mixed with allograft or xenograft, which was significantly larger than that of alloplasts, such as HA or β-TCP (average 2.2 ± 1.2 mm). Furthermore, regarding the volume of newly formed bone, mixed autogenous bone and allograft or xenograft showed a larger defect fill rate (56.6 ± 24.0%) compared to autogenous bone alone (51.5 ± 15.9%), alloplast bone (48.1 ± 6.5%), xenograft alone (45.6 ± 21.4%), and allograft alone (33.2 ± 14.9%); however, the differences were not statistically significant. As can be seen from these results, the combination of resorption–replacing and non-resorptive (residual) materials is effective both histologically and clinically, especially in cases of horizontal bone augmentation.

Additionally, Moy and Aghaloo [60] identified technical risk factors for bone augmentation procedures, which included ① the quality of soft tissues at the augmentation site (influenced by periodontal diseases in adjacent teeth, etc.), ② postoperative inflammation and infection, ③ wound dehiscence owing to the use of removable prosthetics after surgery, ④ technical issues concerning the surgeon, ⑤ poor stabilisation of graft materials, ⑥ blood flow to the graft material, and ⑦ the establishment of a healing period. Therefore, to avoid early implant failure when performing horizontal or vertical bone augmentation, it is crucial not only to assess the bone defect using CBCT images and select appropriate graft materials, but also to consider postoperative management including the use of removable prosthetics to avoid complications, flap design based on the wound healing, the condition of soft tissues at the augmentation site, size of the defect, and establishment of the healing period based on the graft material. Furthermore, in a study on bone augmentation for implants, Aloy-prosper et al. [61] reported survival rates of 95.7% and 97.3%, success rates of 93.6% and 96.2%, and MBL of 0.54 and 0.43 mm with and without bone augmentation, respectively, indicating that not undergoing bone augmentation demonstrated favourable outcomes. In the bone augmentation group, membrane exposure was 8.4%, and all the implants that failed in the bone augmentation group (failure rate of 4.9%) failed before loading. This suggests that avoiding bone augmentation itself through the use of narrow implants, changing the site, or employing angled placement could be considered to avoid early implant failure.

Until now, no study has evaluated the differences in the success and survival rates of vertical and horizontal bone augmentation based on the graft material used. However, similar to ARP, early implant failure is a crucial factor influencing osseointegration. Nevertheless, bone augmentation (horizontal and vertical) in areas where the implant does not make contact does not adversely affect osseointegration. Therefore, it is necessary to separately consider the impact of internal bone defects, such as sinus augmentation, which involves significant thread exposure of the implant and external bone defects.

### 4.3. Sinus Augmentation (Lateral Approach/Crestal Approach)

#### 4.3.1. Relationship between Early Implant Failure and Sinus Augmentation

Barone et al. [62] reported a significantly lower implant survival rate for implants with sinus augmentation (86.1%) compared to those without sinus augmentation (96.4%) in a study comprising 105 cases and 393 implants (Figure 4). All instances of implant failure occurred within one year of implantation. Similarly, Cannizzaro et al. [63] reported that all failed implants during a 5-year observation period were attributed to early implant failure occurring before loading, with postoperative inflammation, infection, and maxillary sinusitis as contributing factors. Additionally, Zinser et al. [64] conducted an investigation into implant loss in 1045 implants of 347 cases undergoing sinus augmentation, revealing a high implant survival rate of 93.3%. However, 80% of the failed implants failed before loading, and the remaining 20% failed within two years of loading. The risk factors for implant failure demonstrated no difference in the implant diameter, length, or surface characteristics. However, smoking (HR: 1.98), one-stage procedures (HR: 2.56), graft material (allograft/xenograft, HR: 4.74), pre-existing bone (0–2 mm, HR: 3.51), and overall poor health status (HR: 2.73) have been reported as significant risk factors. Additionally, Ohayon et al. [65] conducted a study on bone graft material leakage from the window following sinus lift and observed bone graft material leakage into the buccal mucosa at distances ranging from 0 to 12.2 mm (average 3.8 mm) 6 months after sinus lift using xenograft material, suggesting a potential risk of postoperative infection.

Considering the lateral approach and implant failure, Kozuma et al. [66] evaluated the factors contributing to postoperative infection and implant failure in the lateral approach and identified the following risk factors for postoperative infection: ① chronic sinusitis (OR: 16.7) and ② simultaneous surgery, while the risk factors for implant failure included ① chronic sinusitis, ② male sex, ③ diabetes, ④ use of removable dentures after surgery, and ⑤ perforation of the sinus membrane, thus highlighting the importance of host factors (systemic diseases) and postoperative management, such as the use of dentures. Furthermore, Guerrero [67] reported on implantation techniques (whether simultaneous or delayed implantation) and found that among the 11% of cases with early implant failure, 10% were associated with simultaneous implantation and 1% with delayed implantation, indicating a significantly higher rate of early failure with simultaneous procedures. Therefore, sinus augmentation requires careful preoperative diagnosis, selection of surgical techniques, including graft materials, and postoperative management; more so than other bone augmentation procedures. 

On the other hand, regarding the relationship between the crestal approach and early implant failure, a systematic review by Shi et al. [68] investigated the outcomes of the crestal approach using the osteotome technique in 1977 individuals with 3119 implants. They reported that, among the 102 failed implants, 82.4% failed within 1 year after loading, and the application of short implants (<8 mm) significantly lowered the survival rate (83.3% vs. 96.3%). Furthermore, a systematic review and meta-analysis by Calin et al. [69] reported that approximately 60% of implant failures occur before the placement of the prosthesis, and a difference in implant failure exists between cases with an existing bone height of ≥4 mm and those with <4 mm. Furthermore, in a recent report by Li et al. [70], a high 3-year survival rate of 96.9% was reported; however, all instances of early implant failure before loading constituted one-third of all implant failures, and the experience of the surgeon was identified as a high-risk factor (HR: 12.95). This highlights that, similar to the lateral approach, the crestal approach, aimed at minimal invasiveness, is strongly correlated with early implant failure, which underscores the importance of preoperative diagnosis and surgical techniques in this treatment. Additionally, regarding the relationship between complications and early implant failure, Stacchi et al. [71] conducted a study on 430 patients with an existing bone height of ≤5 mm (average 4.0 ± 0.9 mm) who underwent the crestal approach and reported a 2.8% occurrence of early implant failure, with complications including sinus membrane perforation (7.2%), maxillary sinusitis (0.9%), implant displacement within the maxillary sinus (0.7%), and occurrences of dizziness and infection (0.2%). Furthermore, the relationship between the sinus width and membrane perforation in the crestal approach was found to be a more significant risk factor for early implant failure than the surgical technique or smoking, and the study reported that a larger sinus width was associated with a higher likelihood of membrane perforation. Therefore, unlike the lateral approach, in the crestal approach where recovery from membrane perforation is challenging, a strong causal relationship exists between membrane perforation and early implant failure. In addition to the osteotome technique, various techniques have been developed for the crestal approach. However, studies indicate that membrane perforation can occur regardless of the technique used, and a strong causal relationship exists between membrane perforation and the amount of sinus lift. Microscopic confirmation has demonstrated the amount of lift (averaging 4–5 mm) to be a limitation [72,73]. Due to the absence of a method to block membrane perforation from the implant cavity, the crestal approach is considered a minimal interventional approach compared to the lateral approach. Therefore, cases should be selected based not only on the existing bone volume but also on the amount of lift. Acquiring the skills to switch techniques to the lateral approach when needed should be considered to minimise membrane perforation, which is a risk factor for early implant failure.

#### 4.3.2. Selection of the Graft Material in Sinus Augmentation

The selection of graft material for general bone augmentation surgeries typically involves situations where only a small aspect of the implant surroundings comes into contact with the graft material, and the impact of occlusal loads on the graft material itself is relatively small. However, in certain procedures, such as sinus augmentation, ARP, and delayed GBR where the graft material occupies areas beyond the neck of the implant, when selecting the graft material, the occlusal loading conditions imposed by the superstructure placement should be considered. Therefore, understanding the mechanism and amount of new bone formation over what period (how much of the graft material replaces bone) is crucial in these cases.

A systematic review by Pesce et al. [74] revealed: ① the relationship between existing bone volume and NBF is irrelevant to the graft material used; ② higher vertical bone augmentation volumes are associated with lower rates of NBF, a trend particularly pronounced with the use of xenografts and alloplasts.; and ③ the resorption of graft material at 6 months post-transplantation is the lowest in xenografts (7.30%) followed by alloplasts (27.8%). These findings suggest that graft materials with lower resorption rates tend to have higher survival rates. Additionally, larger vertical bone augmentation volumes may lead to insufficient blood supply, including osteoblasts from existing bone and periosteum, resulting in reduced NBF. As mentioned above, sinus augmentation differs from other bone augmentation surgeries in that it has additional load-bearing requirements. Therefore, the use of materials with higher survival rates (low bone replacement rate), such as xenografts and alloplasts, should be avoided in patients with low existing bone volume and large vertical bone augmentation volume.

According to a systematic review by Al-Morassi et al. [75] on graft material osteoconduction in sinus augmentation: ① The NBF rates for alloplasts and xenografts were significantly lower than those for autogenous bone over the entire observation period (3–15 months) with differences of −8.04% and −4.49%, respectively. In the first 6 months (3–5 months), the NBF rates for alloplast alone (−10.66%) and xenograft alone (−7.93%) were significantly lower compared to that of autogenous bone. For the period beyond 6 months (6–15 months), only alloplast alone showed a significantly lower NBF rate (−7.06%) compared to autogenous bone; ② Residual graft material (RG) rates over the entire observation period were significantly higher for xenograft (9.62%) compared to autogenous bone. Furthermore, for the period beyond 6 months, both alloplast alone (12.03%) and xenograft alone (14.62%) showed significantly higher residual rates compared to that of autogenous bone transplantation. Although not statistically significant, allograft alone and allograft + xenograft had lower residual rates (−3.06% and −4.26%, respectively) compared to autogenous bone alone; ③ In terms of treatment ranking, for NBF, the order was allograft + xenograft > autogenous bone alone > autogenous bone + alloplast. Xenograft alone and alloplast alone showed lower NBF rates. In terms of RG rates, the ranking was autogenous bone alone > autogenous bone + alloplast > allograft + xenograft > allograft alone. Alloplast alone and xenograft alone exhibited higher RG rates.

These results indicate an inverse relationship between the RG rate and the NBF rate, suggesting that the residual graft material delays the formation of new bone and that NBF and RG changes greatly depending on the healing period. Therefore, for sinus augmentation, which requires occlusal loading conditions, stricter control over the selection of graft material and establishment of the healing period than that for other bone augmentation procedures (especially IIP and alveolar ridge augmentation simultaneously with implant placement) will play a major role in preventing early implant failure.

In particular, when using xenografts alone, recent long-term follow-up cases of sinus augmentation have reported cases of maxillary sinusitis-associated peri-implantitis, triggered by peri-implantitis around the implant. These cases were reported in case series of Parks et al. [76] and Scarano et al. [77]. Thus, xenografts should be avoided in sinus augmentation with limited pre-existing bone volume.

## 5. Characteristics and Selection of Various Bone Grafting Materials

Bone regeneration is performed to increase the bone volume through bone augmentation, as mentioned above. The patient’s autogenous bone is used for grafting; however, due to the invasive nature of the procedure at the bone harvesting site and the limited amount of bone that can be harvested, allograft or alloplast is used instead. These are called graft materials. Graft materials are intended to provide a scaffold and must have a surface on which cells can migrate, adhere, proliferate, and differentiate. Porous powders or blocks are used as graft materials. In addition, a cell-blocking membrane should be used to physically block the connective tissue-derived cells to preserve space. Bone defects, which are the sites of healing, can be regenerated with bone tissue by stimulating the proliferation and differentiation of osteoblasts to form bone.

Bone graft materials are classified into autogenous bone, allografts, and alloplasts. Autogenous bone has the highest bone regeneration capacity; however, the amount of bone that can be harvested is limited, and secondary procedures are required for harvesting. Autogenous bone can be categorised into homologous and xenogeneic autogenous bones. Homologous autogenous bone is derived from human donors, subjected to decalcification and freeze-drying processes, possessing both osteoinductive and osteoconductive properties, making them superior in terms of bone regeneration. However, their use is limited due to ethical considerations. Xenogeneic autogenous bone derived from bovine sources is widely used in dental implant surgery worldwide. Bovine bones are deproteinated by high-temperature calcination, reduced to inorganic components, and then mechanically crushed into granules. Electron microscopy reveals variations in the pore sizes on the surface, ranging from open to less open. Composition analysis shows the presence of HA. When transplanted, they partially dissolve at the implantation site, promoting the formation of new bone in the surrounding area. Calcium phosphate, an artificial bone, comes in various types and is used in clinical applications. It exhibits excellent osteoconductivity but has low hydration reactivity and lacks self-hardening properties. HA, β-TCP, and octacalcium phosphate are used as graft materials. Electron microscopy images show that artificial materials are designed with standardised pores. HA is used in granular or porous block forms. The strength and osteoconductivity of HA are influenced by the crystallinity during synthesis. Generally, higher crystallinity results in higher strength but poorer osteoconductivity. Although HA has excellent osteoconductivity, its granules remain intact as they are not absorbed within the bone. β-TCP is used in granular form and has greater solubility than HA; it is absorbed within the bone and easily replaced by new bone [78].

When clinically applying bone augmentation procedures to implants, the selection of the graft material should be considered in terms of bone substitution. Non-resorbable xenografts (especially bovine bone) show lower NBF values compared to graft materials other than autogenous bone; thus, although small-scale thread exposure of approximately 2–3 mm may not cause issues, in cases with a larger thread exposure, alveolar ridge augmentation during the healing period, or in areas subject to load conditions, such as sinus augmentation, careful consideration of graft material adaptation, particularly avoiding the sole use of non-resorbable xenografts, and adjusting the healing period are crucial in bone augmentation procedures for avoiding early implant failure.

## 6. Conclusions

Upon reviewing the risk factors for various bone augmentation techniques and the selection of graft material to avoid early implant failure, we found that the choice of graft material should be based on pathological NBF volume and RG rates rather than on clinical outcomes assessed through imaging techniques, such as CBCT. Non-resorbable xenografts, particularly bovine bone, exhibit lower NBF values compared to graft materials other than autogenous bone. Therefore, for small-scale thread exposure of approximately 2–3 mm, significant issues may not arise. However, xenograft should not be used alone for graft material in areas subjected to loading conditions, such as ARP, alveolar ridge augmentation during waiting, cases with large thread exposure, and sinus augmentation. 

Although we could not establish appropriate bone graft material in this review, it is very important to understand that the NBF volume and RG rate change depending on the bone graft material and the healing period. Further, selecting an appropriate bone graft material for various bone grafting procedures is paramount to avoid early implant failure.

In addition, new materials, such as carbonate apatite, calcium carbonate, and calcium phosphate are being considered for artificial bone; thus, future studies should focus on understanding the characteristics of each material before using them.

## Figures and Tables

**Figure 1 bioengineering-11-00192-f001:**
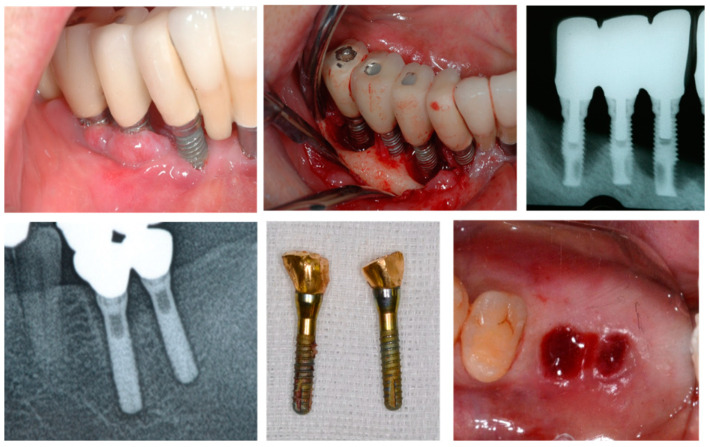
Late implant failure. The (**top row**) shows peri-implantitis and the (**bottom row**) shows disintegration attributed to overload. Late failure differs from early failure as it occurs over time, and its causes and diagnostic and treatment methods have been established.

**Figure 2 bioengineering-11-00192-f002:**
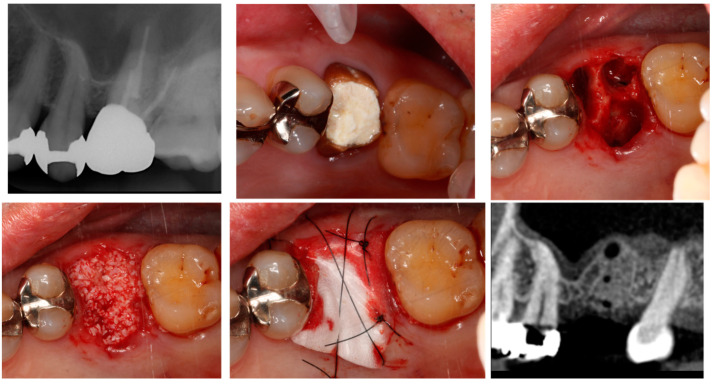
Alveolar ridge preservation.

**Figure 3 bioengineering-11-00192-f003:**
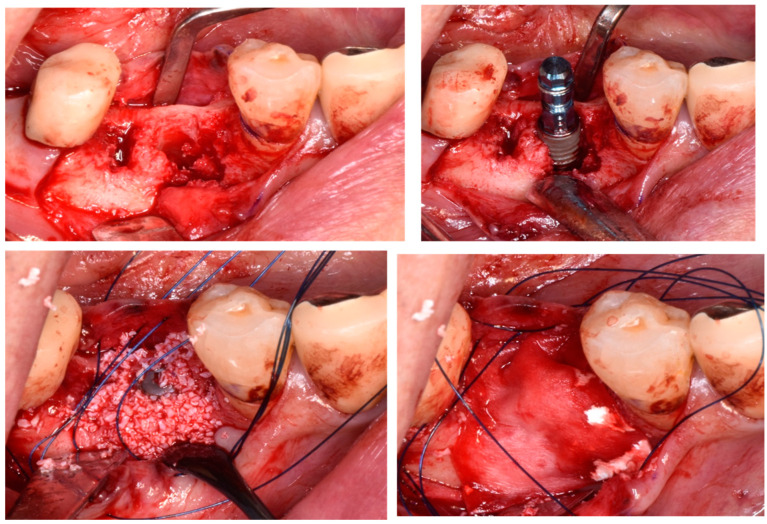
Alveolar ridge augmentation.

**Figure 4 bioengineering-11-00192-f004:**
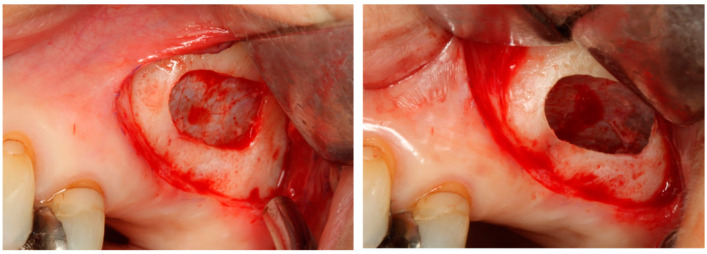
Sinus augmentation (lateral approach).

**Table 1 bioengineering-11-00192-t001:** Definition and risk factors for early implant failure.

Study	Patients/ Implant	Definition of Early Failure	Implant Placement Year	Yes	No	Rate of Early Failure
**Olmedo-Gaya et al., 2016** [20]	142/276	Occurs before loading	2007–2011	Male, severe periodontal disease, short implant (7–8.5 mm), bone augmentation, pain and inflammation 1 week postoperatively	Age, systemic disease, smoking, alcohol consumption, bruxism, edentulous jaw, implant site and diameter, bone quality, bone augmentation	Implant level: 5.8%
**Chrcanovic et al.,** **2016** [21]	2670/10,096	Before abutment connection	1980–2014	Smoking, antidepressants	Age, sex, bruxism, systemic disease, irradiation, hormone replacement therapy, antiplatelet drugs, immunosuppressants	Implant level: 6.36%
**Grisar et al.,** **2017** [22]	509/1139	Occurs before loading	2012–2014	Male, smoking, edentulous jaw	Age, alcohol abuse, radiation	
**Lin et al.,** **2018** [5]	18,199/30,959	Before abutment connection	2011–2015	Male, elderly, lower anterior teeth, bone augmentation (OR, 1.29)	Number, diameter and length of implants	Failure rate within 1 year: 38.8% failure
**Camps-Font et al.,** **2018** [23]	1322/2673	Before prosthetics placement	2004–2015	Rough-surfaced coloured implants, mandible	Sex, ASA classification, smoking, type of periodontal disease, implant system	Implant level: 1.38% Patient level: 2.80%
**Kang et al.,** **2019** [24]	409/1031	Before or within a few weeks after placement of final superstructure	2015–2017	Mandible, experience of surgeon	Sex, age, diameter and length of implants, type of maxillary sinus floor elevation, bone augmentation	Implant level: 4.1% (of which, early failure 3.3%) Patient level: early 6.5%, Late 1.7%
**Borba et al.,** **2017** [25]	202/774	Occurs before placement of provisional restoration	2002–2014	Bone augmentation (OR, 2.7)	Age, sex, site, implant diameter/length	Implant level: 3.2% Patient level: 8.9%
**Hirota et al.,** **2018** [26]	219/563	Occurs before loading	2005–2017	Postoperative wound dehiscence, optimal functionalisation	Surface properties, bone quality	Optimal functionalisation reduces early failure Implant level: 2.7%
**Chang** **2020** [27]	376/1050	Occurs before placement of final superstructure	2003–2016	Bone augmentation (OR, 9.45), surgical technique including skills and experience		Patient level: 4.8% Implant level: 4.7%
**De Angelis et al.,** **2017** [28]	272/871	Occurs before placement of final superstructure	1998–2006	Bruxism, smoking	Age, sex, implant length	Implant level: early 6.8%, late 8.9%
**Jemt T et al.,** **2017** [29]	2848/9582	①Before abutment placement ②Before placement of superstructure ③Up to one year after placement of superstructure	2003–2011	Bone resorption, Both jaws, number of implants, not prosthetic treatment at the referred clinic, surgeon		Case level: ① 1.4% ② 2.1% ③ 2.3%
**Antoun et al.,** **2016** [30]	1017/3080	Placement to one year of loading	2000–2011	Smoking (OR, 2.08), surgical technique (OR, 3.7), simultaneous GBR, immediate tooth extraction (OR, 2.09), one-stage procedure		Implant level: 1.6%; patient level: 4.0%
**Yang et al.,** **2021** [31]	1078/2053	Placement to one year of loading	2006–2017	Bone quality Type Ⅰ (OR, 3.689), placement immediately after tooth extraction (OR, 3.509), implant length < 10 mm (OR, 2.972), male, age (30–60)	Bone augmentation (1.742)	Implant level: 4.0%
**Jemt et al.,** **2017** [32]	2566/14,083	Placement to several weeks after prosthetic placement	1986–1997	Edentulous jaw (11.3%) 60 years or older < 60 years old		209 (71.8%) had implant failure before superstructure placement; 35 (12.0%) had implant failure between superstructure placement to first maintenance
**Tattan et al.,** **2021** [33]	201/	Before prosthetic placement	2008–2019	Socket preservation (HR, 7.5), soft tissue grafting (HR, 5.03), or bone grafting (HR, 3.4) at the same time of implantation	Smoking, diabetes, osteoporosis, history of periodontal disease, implant length and design, type of graft material	Patient level: 30.3%
**Wu et al.,** **2021** [34]	3785/6113	Before prosthetic placement	2015–2019	Maxilla (OR, 3.7): molar (OR, 2.73); implant surface characteristics, bone graft; Mandible: anterior teeth, male, bone graft	Implant length, design, and shape	Patient level: 1.6%; Implant level: 1.2%
**Staedt et al.,** **2020** [35]	/9080	Before abutment connection	2002–2012	Lower molar, young patients	Gender, systemic disease, diabetes	
**Malm et al.,** **2018** [36]	4899/25,781	1 year after superstructure placement	1986–2013	Bone quality, implant surface characteristics, age, number of implants	Gender	Failure in 8.6% of edentulous cases before prosthetic placement (implant level: 1.6%; patient level: 6.3%)
**Carr et al.,** **2019** [37]	362/8540	Within 1 year after placement	1983–2014	Bone grafting alone, ridge preservation, xenograft, postoperative complications	Age, gender, periodontal disease	Implant level: 4.2%

OR: odds ratio; GBR: guided bone regeneration; HR: hazards ratio; ASA: American Society of Anaesthesiologists.

## Data Availability

The datasets used and analysed during the current study are available from the corresponding author on reasonable request.

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
