# Peer review of "Risk Factors for Early Implant Failure and Selection of Bone Grafting Materials for Various Bone Augmentation Procedures: A Narrative Review"

_bioengineering, 2024, doi:10.3390/bioengineering11020192_

Round 1

Reviewer 1 Report (Previous Reviewer 1)

Comments and Suggestions for Authors

Upon reading your text and reviewing the conclusion, I am unable to identify the Optimal Graft Material stated in the title.

The abstract indicates a reliance on reports and systematic reviews to support the statement on the high long-term success of dental implant therapy. However, this information should not be in this section. If you are reporting literature, references should be included. If references cannot be incorporated into the abstract, explanations of previous reports should not be presented in this section.

Topic 2 is named "Early Implant Failure Causes," but there is no corresponding table or paragraph describing these causes. The information is scattered and correlated with other factors in the manuscript, making it challenging to extrapolate more than understanding "what was done" by other authors.

In summary, I observe that the review, while containing accurate information, is presented in a somewhat simplistic manner. It lacks a clear search definition, a structured research methodology, and improvements in data presentation. It is imperative to delineate which factors truly contribute to early osseointegration loss and distinguish them from the effects. There is a need for a more pronounced focus on bone graft materials, given their crucial role as indicated by the title.

Expectations for a narrative review include it serving as a robust study guide for new dental professionals. Despite the apparent length of the text, a closer examination of individual topics reveals a notable lack of substantive depth. Furthermore, the presence of extensive sentences and paragraphs without proper references undermines the credit due to authors who have previously defined or researched the information being presented.

Author Response

Reviewer1 Comments

Thank you very much for your appropriate review. We have corrected and removed the relevant text, as per your suggestions, which is indicated in the revised manuscript with blue highlight.

1.Upon reading your text and reviewing the conclusion, I am unable to identify the Optimal Graft Material stated in the title.

Answer: We agree that you have a good point regarding the content of the manuscript. Since we were unable to identify an appropriate bone graft material in this review, we have revised the content, including the title and conclusions.

2.The abstract indicates a reliance on reports and systematic reviews to support the statement on the high long-term success of dental implant therapy. However, this information should not be in this section. If you are reporting literature, references should be included. If references cannot be incorporated into the abstract, explanations of previous reports should not be presented in this section.

Answer: Thank you for bringing this issue to our attention. We have removed the references from the abstract and modified the text accordingly.

3.Topic 2 is named "Early Implant Failure Causes," but there is no corresponding table or paragraph describing these causes. The information is scattered and correlated with other factors in the manuscript, making it challenging to extrapolate more than understanding "what was done" by other authors.

Answer: We apologise for the confusion in this section and mismatch between the heading and its content. Accordingly, the heading of this section has been changed to ensure it corresponds with its content.

4.In summary, I observe that the review, while containing accurate information, is presented in a somewhat simplistic manner. It lacks a clear search definition, a structured research methodology, and improvements in data presentation. It is imperative to delineate which factors truly contribute to early osseointegration loss and distinguish them from the effects. There is a need for a more pronounced focus on bone graft materials, given their crucial role as indicated by the title.

Answer: Thank you for your careful review of our manuscript, and upon reflection, we agree that its content is scattered and not in line with the title. To this end, we have removed all text that we considered irrelevant and not in line with the purpose of this manuscript.

5.Expectations for a narrative review include it serving as a robust study guide for new dental professionals. Despite the apparent length of the text, a closer examination of individual topics reveals a notable lack of substantive depth. Furthermore, the presence of extensive sentences and paragraphs without proper references undermines the credit due to authors who have previously defined or researched the information being presented.

Answer: Thank you for your suggestion. We have modified the manuscript to present a review of knowledge regarding early implant failure and bone graft material selection, serving as a study guide. In addition, we have removed sentences that contained personal opinions without supporting references.

Reviewer 2 Report (Previous Reviewer 3)

Comments and Suggestions for Authors

Dear Authors,

The manuscript has improved. However, I think that some further modifications are still required

Regarding the previous review:

1) I suggest to include a paragraph regarding new transmucosal neck convergent neck and hyperbolic neck implants.. A search strategy in WOS using keywords as “hyperbolic neck implants” convergent neck implants”, transmucosal implants could be useful for the purpose. Different surgical approaches were proposed from research groups (subcrestal insertion, tissue level insertion) and could results in different peri implant complications.

Answer: Thank you for your comment. As the reviewer pointed out, transmucosal convergent neck and hyperbolic neck implants are an interesting topic that is likely to become very relevant in the near future. Unfortunately, we as authors do not feel we have the required level of expertise on this topic to include it as part of our review article. In addition, we could only find a limited amount of studies on these types of implants when we performed a search in the PubMed database, suggesting that they are not part of a generally recognized category. Therefore, we have decided not to add additional information on this particular topic to the revised manuscript. We hope that you will still find the revised manuscript worthy of being considered for publication.

I report here some articles useful to be included in the manuscript. Transmucosal implants are gaining higher interest towards clinicians as useful to avoid second stage surgeries. This protocol is also useful for cement retained restorations as the implant abutment connection generally is located above the soft tissues. This will reduce the cement retained restoration complications and failures. Some short medium-term studies are published in this context.

Ceruso FM, Ieria I, Tallarico M, Meloni SM, Lumbau AI, Mastroianni A, Zotti A, Gargari M. Comparison between Early Loaded Single Implants with Internal Conical Connection or Implants with Transmucosal Neck Design: A Non-Randomized Controlled Trial with 1-Year Clinical, Aesthetics, and Radiographic Evaluation. Materials (Basel). 2022 Jan 10;15(2):511. doi: 10.3390/ma15020511. PMID: 35057240; PMCID: PMC8779815.

Prati C, Zamparini F, Canullo L, Pirani C, Botticelli D, Gandolfi MG. Factors Affecting Soft and Hard Tissues Around Two-Piece Transmucosal Implants: A 3-Year Prospective Cohort Study. Int J Oral Maxillofac Implants. 2020 Sep/Oct;35(5):1022-1036. doi: 10.11607/jomi.7778. PMID: 32991655.

4)Regarding early complications, Ti ions leakage has been confirmed as a potential cause of early lack of integration (and failure). A number of histological and ESEM-EDX investigation reported the occurrence of this event, with different ourcomes when comparing mandible, maxilla and auigmented sites. Please include a paragraph regarding this controversial aspect.

Answer: As the reviewer pointed out, leakage of titanium ions as a potential cause of early implant failure is quite interesting as a topic for discussion. However, we could not find any scientific studies focusing on this phenomenon after performing a search in the PubMed database, and therefore it does not seem to be generally confirmed in the field of implant dentistry. For the time being, we would prefer not to include this issue in our manuscript. If the reviewer could mention specific studies that are relevant for the discussion of this topic, we would greatly appreciate it.

Response. Regarding Ti ions leakage, I suggest to include this paper. Controversial data exists in literature, some data reports no histological findings between Ti remnants in bone, others suggest a critical role in bone remodeling.

Some articles could be useful to be added in the discussion.

Asa'ad F, Thomsen P, Kunrath MF. The Role of Titanium Particles and Ions in the Pathogenesis of Peri-Implantitis. J Bone Metab. 2022 Aug;29(3):145-154. doi: 10.11005/jbm.2022.29.3.145. Epub 2022 Aug 31. PMID: 36153850; PMCID: PMC9511127.

Comments on the Quality of English Language

english is fine

Author Response

Reviewer 2 Comments

Thank you very much for your appropriate review. We have corrected and removed the relevant text, as per your suggestions, which is indicated in the revised manuscript with blue highlight.

The manuscript has improved. However, I think that some further modifications are still required

Regarding the previous review:

1) I suggest to include a paragraph regarding new transmucosal neck convergent neck and hyperbolic neck implants.. A search strategy in WOS using keywords as “hyperbolic neck implants” convergent neck implants”, transmucosal implants could be useful for the purpose. Different surgical approaches were proposed from research groups (subcrestal insertion, tissue level insertion) and could results in different peri implant complications.

Answer: Thank you for your comment. As you have highlighted, transmucosal convergent neck and hyperbolic neck implants are interesting topics, likely to become very relevant in the near future. Unfortunately, we as authors do not feel we have the required level of expertise on this topic to include it as part of our review article. In addition, we could only find a limited number of studies on these types of implants when we performed a search of the PubMed database, suggesting that they are not yet part of a generally recognised category. Therefore, we have decided not to add additional information on this particular topic to the revised manuscript. We hope that you will still find the revised manuscript worthy of being considered for publication.

I report here some articles useful to be included in the manuscript. Transmucosal implants are gaining higher interest towards clinicians as useful to avoid second stage surgeries. This protocol is also useful for cement retained restorations as the implant abutment connection generally is located above the soft tissues. This will reduce the cement retained restoration complications and failures. Some short medium-term studies are published in this context.

Ceruso FM, Ieria I, Tallarico M, Meloni SM, Lumbau AI, Mastroianni A, Zotti A, Gargari M. Comparison between Early Loaded Single Implants with Internal Conical Connection or Implants with Transmucosal Neck Design: A Non-Randomized Controlled Trial with 1-Year Clinical, Aesthetics, and Radiographic Evaluation. Materials (Basel). 2022 Jan 10;15(2):511. doi: 10.3390/ma15020511. PMID: 35057240; PMCID: PMC8779815.

Prati C, Zamparini F, Canullo L, Pirani C, Botticelli D, Gandolfi MG. Factors Affecting Soft and Hard Tissues Around Two-Piece Transmucosal Implants: A 3-Year Prospective Cohort Study. Int J Oral Maxillofac Implants. 2020 Sep/Oct;35(5):1022-1036. doi: 10.11607/jomi.7778. PMID: 32991655.

Answer: According to the reviewer’s suggestion, the revised manuscript discusses implant designs along with the recommended supporting references.

4)Regarding early complications, Ti ions leakage has been confirmed as a potential cause of early lack of integration (and failure). A number of histological and ESEM-EDX investigation reported the occurrence of this event, with different outcomes when comparing mandible, maxilla and augmented sites. Please include a paragraph regarding this controversial aspect.

Answer: As you have pointed out, leakage of titanium ions as a potential cause of early implant failure is quite interesting as a topic for discussion. However, we could not find any scientific studies focusing on this phenomenon after performing a search of the PubMed database, and therefore it does not seem to be generally confirmed in the field of implant dentistry. For the time being, we would prefer not to include this issue in our manuscript. If the reviewer could mention specific studies that are relevant for the discussion of this topic, we would greatly appreciate it.

Response. Regarding Ti ions leakage, I suggest to include this paper. Controversial data exists in literature, some data reports no histological findings between Ti remnants in bone, others suggest a critical role in bone remodeling.

Some articles could be useful to be added in the discussion.

Asa'ad F, Thomsen P, Kunrath MF. The Role of Titanium Particles and Ions in the Pathogenesis of Peri-Implantitis. J Bone Metab. 2022 Aug;29(3):145-154. doi: 10.11005/jbm.2022.29.3.145. Epub 2022 Aug 31. PMID: 36153850; PMCID: PMC9511127.

Answer: Thank you for your valuable insight; accordingly, the revised manuscript acknowledges the role of Ti ion leakage along with the recommended supporting reference.

Round 2

Reviewer 1 Report (Previous Reviewer 1)

Comments and Suggestions for Authors

The authors updated the manuscript accordingly.

Reviewer 2 Report (Previous Reviewer 3)

Comments and Suggestions for Authors

the paper has improved. It can be accepted now. Please check the text for uniform formatting.

This manuscript is a resubmission of an earlier submission. The following is a list of the peer review reports and author responses from that submission.

Round 1

Reviewer 1 Report

Comments and Suggestions for Authors

According to the authors, the Institutional Review Board Statement was not applicable, and the Informed Consent Statement was also not applicable. However, there are several photos from patients in the text. How can that be?

For a narrative review, I missed more references in the lines. This is a scientific article, not a book. You must give credit to the previous authors who introduced the concepts instead of just writing long paragraphs without citing anyone and taking over the credit for it.

Abstract: The abstract could benefit from a clearer structure that introduces the main points in a more organized manner. Start with the significance of implant therapy, briefly mention the types of failures, and then delve into the factors causing early implant failure. Also, state the specific objectives or aims of the review more clearly. What is the ultimate goal or purpose of reviewing these bone augmentation techniques and associated risk factors?

Keywords: Please update the keywords following MeSH (Medical Subject Headings).

Introduction: The first paragraph in your introduction is excessively long and contains sentences that could benefit from being split into shorter, more digestible segments. Additionally, strengthening your conceptual information, such as the 'survival rate,' by incorporating more references to support these assertions can enhance the credibility and depth of your content.

How was Figure 1 obtained? What kind of data was used, and what is the original source?

Lines 63 to 71 in Topic 2.1 require references.

Are the clinical photos from the authors or previous articles? Provide the source in the figure legends and include an ethical approval statement at the end of the paper to justify the inclusion of clinical images from patients here.

Page 3, lines 78-87 need references.

How is abfraction (Figure 3) related to early implant failure?

The title of Topic 2.2 is too generic.

Standardize whether the table’s title will be placed above or below each table.

Table 1 seems unnecessary; the information could be better integrated into the text.

In Table 3, the inclusion of Sex and Gender is confusing. Are you considering both genders? Explain how Sex can be a host factor for implant failure.

Table 3 seems subjective and empiric rather than scientific. It's unclear how systemic diseases are categorized. There's an overlap between medications prescribed daily under the host factor column and Patient Management (antimicrobial administration and use of removable dentures) under Surgeon-related factors. Consider removing Table 3 altogether

Author Response

Please note that revised text has been highlighted in yellow.

1.According to the authors, the Institutional Review Board Statement was not applicable, and the Informed Consent Statement was also not applicable. However, there are several photos from patients in the text. How can that be?

Answer: Thank you for the comment. Regarding the informed consent required for the inclusion of the case photographs as part of the manuscript, we have now included a specific statement at the end of the article declaring that informed consent was obtained from the patients..

2.For a narrative review, I missed more references in the lines. This is a scientific article, not a book. You must give credit to the previous authors who introduced the concepts instead of just writing long paragraphs without citing anyone and taking over the credit for it.

Answer: Thank you for your comment. The relevant references have been added to the revised text. We hope that all the statements included in the manuscript are now appropriately supported by references to the original studies.

3.Abstract: The abstract could benefit from a clearer structure that introduces the main points in a more organized manner. Start with the significance of implant therapy, briefly mention the types of failures, and then delve into the factors causing early implant failure. Also, state the specific objectives or aims of the review more clearly. What is the ultimate goal or purpose of reviewing these bone augmentation techniques and associated risk factors?

Answer: Thank you for the suggestion. The abstract has been revised in a way that describes the main points and objectives more clearly.

4.Keywords: Please update the keywords following MeSH (Medical Subject Headings).

Answer: Thank you for the suggestion. We have updated the keywords, which now follow MeSH.

5.Introduction: The first paragraph in your introduction is excessively long and contains sentences that could benefit from being split into shorter, more digestible segments. Additionally, strengthening your conceptual information, such as the 'survival rate,' by incorporating more references to support these assertions can enhance the credibility and depth of your content.

Answer: Thank you for your suggestions. The Introduction section has been shortened in the revised manuscript, the appropriate references have been added, and the revisions that you suggested have been implemented. We hope that the revised Introduction section is more adequate and easier to follow.

6.How was Figure 1 obtained? What kind of data was used, and what is the original source?

Answer: Thank you for your question. We have decided to remove Figure 1 from the revised manuscript because we found that it did not describe the characteristics of early implant failure in an adequate manner.

7.Lines 63 to 71 in Topic 2.1 require references.

Answer: Thank you for the suggestion. The relevant references have been added to the revised text.

8.Are the clinical photos from the authors or previous articles? Provide the source in the figure legends and include an ethical approval statement at the end of the paper to justify the inclusion of clinical images from patients here.

Answer: Thank you for the suggestion. As the clinical photographs were taken by the authors, a statement declaring that written informed consent from the patients was obtained for their inclusion in the manuscript has now been added at the end of the article.

9.Page 3, lines 78-87 need references.

Answer: Thank you for the suggestion. The relevant references have been added to the revised text.

10.How is abfraction (Figure 3) related to early implant failure?

Answer: Thank you for your question. Since we have realized that Figure 3 was not necessary for this article, it has been removed from the revision.

11.The title of Topic 2.2 is too generic.

Answer: Thank you for your comment. The title of topic 2.2 has therefore been revised following your comment.

12.Standardize whether the table’s title will be placed above or below each table.

Answer: Thank you for the suggestion. The position of the table titles has been made consistent throughout the revised manuscript. We apologize for this formatting oversight in the original manuscript. .

13.Table 1 seems unnecessary; the information could be better integrated into the text.

Answer: Thank you for the suggestion. We have removed the original Table 1 from the revised manuscript, as we agree with the reviewer in that it was unnecessary.

14.In Table 3, the inclusion of Sex and Gender is confusing. Are you considering both genders? Explain how Sex can be a host factor for implant failure.

15.Table 3 seems subjective and empiric rather than scientific. It's unclear how systemic diseases are categorized. There's an overlap between medications prescribed daily under the host factor column and Patient Management (antimicrobial administration and use of removable dentures) under Surgeon-related factors. Consider removing Table 3 altogether

Answer: Thank you for the suggestion. Table 3 has been removed from the revised manuscript following your suggestion, as we agreed with the criticism provided.

Reviewer 2 Report

Comments and Suggestions for Authors

Dear Authors,

I read your manuscript, A narrative review of the Risk Factors for Early Implant Failure 2 and Optimal Graft Material for Bone Augmentation Procedures that can be considered interesting for the readers.

Unfortunately, the manuscript presents some leaks in its structure, among them the total absence of the methods section.

Without a proper methods section, it is impossible to evaluate the quality of the manuscript and the methodology of the review.

I suggest resubmitting the manuscript, after including a methods section with the methodology and data source used in this review and after rechecked the whole manuscript.

Author Response

I read your manuscript, A narrative review of the Risk Factors for Early Implant Failure 2 and Optimal Graft Material for Bone Augmentation Procedures that can be considered interesting for the readers.

Unfortunately, the manuscript presents some leaks in its structure, among them the total absence of the methods section.

Without a proper methods section, it is impossible to evaluate the quality of the manuscript and the methodology of the review.

I suggest resubmitting the manuscript, after including a methods section with the methodology and data source used in this review and after rechecked the whole manuscript.

Answer: Thank you for your valuable feedback. Since this manuscript is a narrative review, it is standard practice to directly follow the Introduction with the relevant contents, without including a separate methodology section. In fact, the Guidelines for Authors provided by Bioengineering specifically state that in the case of reviews, “the structure can include an Abstract, Keywords, Introduction, Relevant Sections, Discussion, Conclusions, and Future Directions”. Please also refer to previous review articles published in the journal Periodontology 2000 (https://onlinelibrary.wiley.com/journal/16000757) for additional examples of this structure.

Reviewer 3 Report

Comments and Suggestions for Authors

Authors made an extensive work in determining the risk factors in early and delayed failures

  I recommend some modifications as it need to improve the manuscript completeness.  

1)      I suggest to include a paragraph regarding new transmucosal neck convergent neck and hyperbolic neck implants..  A search strategy in WOS  using  keywords as “hyperbolic neck implants” convergent neck implants”, transmucosal implants could be useful for the purpose. Different surgical approaches were proposed from research groups (subcrestal insertion, tissue level insertion) and could results in different peri implant complications.

2) Morevoer, authors should better focus on bone levels modifications (Crestal bone loss, or Maginal bone loss). A fast bone loss in the early stages from integration could lead to an early or delayed failures. Clearly, a lower resolution could be expected for this investigations when compared to CBCT. However, it  is more ethically acceptable due to the low dose radiation provided.

3) the authors in the second part of the review reported a wide selection of images and information regarding the bone graft materials. This should be markedly reduced as the primary aim of the study was to analyse the early and late failures. Reviewing of past and new bone regenerative materials seems out of the scope of the article.    

4)Regarding early complications, Ti ions leakage has been confirmed as  a potential cause of early lack of integration (and failure). A number of histological and ESEM-EDX investigation reported the occurrence of this event, with different ourcomes when comparing mandible, maxilla and auigmented sites. Please include a paragraph regarding this controversial aspect.

Comments on the Quality of English Language

English need a minor revision

Author Response

 I recommend some modifications as it need to improve the manuscript completeness. 

1) I suggest to include a paragraph regarding new transmucosal neck convergent neck and hyperbolic neck implants..  A search strategy in WOS using  keywords as “hyperbolic neck implants” convergent neck implants”, transmucosal implants could be useful for the purpose. Different surgical approaches were proposed from research groups (subcrestal insertion, tissue level insertion) and could results in different peri implant complications.

Answer: Thank you for your comment. As the reviewer pointed out, transmucosal convergent neck and hyperbolic neck implants are an interesting topic that is likely to become very relevant in the near future. Unfortunately, we as authors do not feel we have the required level of expertise on this topic to include it as part of our review article. In addition, we could only find a limited amount of studies on these types of implants when we performed a search in the PubMed database, suggesting that they are not part of a generally recognized category. Therefore, we have decided not to add additional information on this particular topic to the revised manuscript. We hope that you will still find the revised manuscript worthy of being considered for publication.

2) Morevoer, authors should better focus on bone levels modifications (Crestal bone loss, or Maginal bone loss). A fast bone loss in the early stages from integration could lead to an early or delayed failures. Clearly, a lower resolution could be expected for this investigations when compared to CBCT. However, it is more ethically acceptable due to the low dose radiation provided.

Answer: As the reviewer pointed out, marginal bone loss is one of the most important aspects for long-term implant success. However, the scope of this review article is limited to early implant failure, which is not necessarily associated with changes in marginal bone levels. Therefore, we have decided not to include this topic on our manuscript this time.

3) the authors in the second part of the review reported a wide selection of images and information regarding the bone graft materials. This should be markedly reduced as the primary aim of the study was to analyse the early and late failures. Reviewing of past and new bone regenerative materials seems out of the scope of the article.   

Answer: Thank you for the suggestion. The manuscript has been revised following your comment, and the unnecessary content has been removed from it.

4)Regarding early complications, Ti ions leakage has been confirmed as a potential cause of early lack of integration (and failure). A number of histological and ESEM-EDX investigation reported the occurrence of this event, with different ourcomes when comparing mandible, maxilla and auigmented sites. Please include a paragraph regarding this controversial aspect.

Answer: As the reviewer pointed out, leakage of titanium ions as a potential cause of early implant failure is quite interesting as a topic for discussion. However, we could not find any scientific studies focusing on this phenomenon after performing a search in the PubMed database, and therefore it does not seem to be generally confirmed in the field of implant dentistry. For the time being, we would prefer not to include this issue in our manuscript. If the reviewer could mention specific studies that are relevant for the discussion of this topic, we would greatly appreciate it.